# Identification of Susceptibility Genes for *Fusarium oxysporum* in Cucumber via Comparative Proteomic Analysis

**DOI:** 10.3390/genes12111781

**Published:** 2021-11-10

**Authors:** Jun Xu, Ke Wang, Qianqian Xian, Ningyuan Zhang, Jingping Dong, Xuehao Chen

**Affiliations:** 1School of Horticulture and Plant Protection, Yangzhou University, Yangzhou 229005, China; 006963@yzu.edu.cn (J.X.); WK19825301461@163.com (K.W.); QQxian1996@163.com (Q.X.); ningyuan1021@163.com (N.Z.); jpdong@yzu.edu.cn (J.D.); 2State Key Laboratory of Vegetable Germplasm Innovation, Tianjin 300192, China

**Keywords:** cucumber, protein sequencing, *Fusarium* wilt, susceptibility genes

## Abstract

*Fusarium* wilt (*FW*) in cucumber (*Cucumis sativus* L.), caused by *Fusarium oxysporum* f. sp. cucumerinum (Foc), poses a major threat to cucumber growth and productivity. However, lack of available natural resistance resources for *FW* restricts the breeding of resistant cultivars via conventional approaches. Susceptibility (S) genes in susceptible host plants facilitate infection by the pathogen and contribute to susceptibility. Loss of function of these S genes might provide broad-spectrum and durable disease resistance. Here, we screened S genes via comparative proteomic analysis between cucumber cultivars Rijiecheng and Superina, which exhibited resistance and high -susceptibility to *FW*, respectively. We identified 210 and 243 differentially regulated proteins (DRPs) in the Rijiecheng and Superina, respectively, and further found that 32 DRPs were predominantly expressed in Superina and significantly up-regulated after Foc inoculation. Expression verification found that *TMEM115* (*CsaV3_5G025750*), encoding a transmembrane protein, *TET8* (*CsaV3_2G007840*), encoding function as a tetraspanin, *TPS10* (*CsaV3_2G017980*) encoding a terpene synthase, and *MGT2* (*CsaV3_7G006660*), encoding a glycosyltransferase, were significantly induced in both cultivars after Foc infection but were induced to a higher expression level in Superina. These candidate genes might act as negative regulators of *FW* resistance in cucumber and provide effective *FW*-susceptibility gene resources for improving cucumber *FW* resistance through breeding programs.

## 1. Introduction

Cucumber (*C. sativus* L.) is one of the most economically important vegetable crops cultivated worldwide [1]. Cucumber *Fusarium* wilt (*FW*) is a typical soil-borne and destructive fungal disease caused by *F.oxysporum* f. sp. cucumerinum (Foc) [2], which severely restricts cucumber growth and yield [3,4,5,6].

When the Foc pathogen successfully invades the plant it wilts of the leaves or even the entire plant, often leading to plant death several days or weeks after infection; thus, it is usually considered as the most severe biotic factor limiting global cucumber production [7]. The disease *FW* is extremely difficult to control because Foc can survive in the soil, straw or seeds for many years or even decades instigating new disease cycles over the long term [8]. Changes in the pathogenicity of Foc have led to the ineffectiveness of certain resistant varieties or fungicides [9,10]. Until now, the related mechanisms of *FW* suppression are largely unknown. There are few methods to effectively control the harm of *FW* including the grafting with rootstock [11], chemical compounds [12,13], and biocontrol via other beneficial fungi or bacteria [14]. However, the most sustainable way to control *FW* disease is to grow resistant cultivars. However, to date, the inheritance of cucumber against *FW* remains poorly understood. The inheritance of *FW* resistance plays important roles in developing resistant breeding resources and varieties as well as identification resistant genes or effect quantitative trait loci (QTLs), and hence, the diverse resources make the results inconsistent [15,16]. For instance, Vakalounakis (2015) described that cucumber resistance to *FW* was affected by a single gene, whereas other researchers found that cucumber *FW* resistance was regulated by multiple genes [17,18]. The QTL *fw2.1* was detected as 1.91-Mb-long region of chromosome 2 by bulkedsegregant analysis (BSA), and further conformed that *fw2.1* was 0.60 Mb contained 80 candidate genes via the fine-map with five insertion/deletion markers [19]. Although efforts have been made to generate wilt resistant cucumber cultivars through traditional breeding, however, available natural resistance resources for *FW* are very limited, meaning that breeding resistant cultivars through conventional approaches is still a challenge. Therefore, to alleviate the negative impact of *FW* disease, a continuous search for alternative control strategies is an urgent requirement.

Accumulating evidence indicates that host plant susceptibility (S) genes facilitate pathogen infection and contribute to susceptibility [20]. Thus, S genes are required for successful pathogen infection, and are considered essential for compatible plant-pathogen interactions. Loss of function of these S genes (which are recessive alleles) might interfere with the compatibility between the host and pathogen, and consequently provide broad-spectrum and durable disease resistance. For instance, *Mildew*
*Resistance*
*Locus*
*O* (*Mlo*) is one of the best-known S genes and encodes a transmembrane protein and is also a prominent example of robustness in durable pathogen-resistance programs. However, instances of disruption of *Mlo* function have been found (as natural mutants) or generated (by induced mutagenesis, gene silencing, and targeted or non-targeted gene knock-out) in several plant species, including wheat [21,22,23], barley [24,25], tomato [26], grape [27,28], and cucumber [29], and significantly improve powdery mildew (PM) resistance. Another wheat PM-susceptibility gene, enhanced disease resistance 1 (*EDR1*), has been targeted using the CRISPR/Cas9 approach, and shown to enhance PM resistance in wheat significantly [30].

Citrus canker is one of the most destructive citrus diseases, causing severe yield losses worldwide [31]. Mutation of the S gene for citrus canker, LATERAL ORGANBOUNDARIES 1 (*CsLOB1*), which plays a critical role in promoting pathogen growth and erumpent pustule formation, using CRISPR-Cas9-mediated promoter editing has conferred high resistance against citrus canker [32]. Recently, two independent studies have shown that disrupting eIF4Es with the CRISPR-Cas9 system can confer resistance to several viruses in *Arabidopsis* [33] and cucumber [34]. Taken together, these studies illustrate the efficacy of genome editing methods in the development of disease-resistant crop varieties by introducing site-specific mutations to disrupt S genes in a transgene-free manner. Currently, genetic resources for *FW* resistance in cucumber are very limited and *FW* has not been effectively controlled by breeding resistant varieties using conventional approaches.

In susceptible host plants, the S genes are usually targeted and induced by pathogens, and they then negatively regulate host resistance. Thus, a cost-effective and environmentally friendly strategy for limiting diseases disrupt S genes to obtain disease-resistant cultivars sustainably. In this study, we performed a comparative proteomic analysis on the Mock and Foc-inoculated roots of two cucumber cultivars, Superina and Rijiecheng, which exhibited high-susceptibility and resistance to *FW*, respectively, to identify the S genes in cucumber defense. We screened 210 and 243 differentially regulated proteins (DRPs) in Rijiecheng and Superina, and further found that 32 DRPs were significantly up-regulated with higher expression levels in Superina after Foc inoculation. Combined with expression verification, we determined that the genes corresponding to proteins TMEM115, TET8, TPS10, and MGT2 were significantly induced in both cucumber cultivars after Foc infection but were induced to a greater expression level in the susceptible cultivar Superina, speculated that these candidate genes might act as negative regulators of *FW* resistance in cucumber. These findings provide important gene resources to highlight the molecular processes of disease susceptibility in cucumber and facilitate further studies to improve resistance to *FW* in cucumber and other *FW*-susceptible host crops.

## 2. Materials and Methods

### 2.1. Plant Materials and Foc Inoculation

Cucumber cultivars ‘Superina’, and ‘Rijiecheng’, which exhibited high susceptibility and resistance to Foc, respectively, were used in this study. The seeds were germinated on wet gauze in a Petri dish at 28 °C in darkness overnight, then the resulting seedlings were grown in a growth chamber with 16 h/8 h day/night at 25 °C/18 °C, respectively.

The Foc acted as a typical soil-borne and destructive fungal disease, which firstly infected the roots of cucumber growth process in field and induced the responsive genes to defense *FW*. We used the cucumber seedlings to inoculate with Foc using the dip-inoculation method [6]. The Foc strain was propagated on potato dextrose agar (PDA) medium-containing plates at 28 °C for 4 d, then cultured in potato dextrose broth (PDB) medium on a shaker at 180 rpm at 28 °C for 3 d. The spore suspension was adjusted to *10^6* spores/mL using sterile distilled water for inoculation of the cucumber seedlings. Seedlings treated with sterile water were used as mock inoculation controls. The seedling roots were harvested, with three repeats, at different time points (0, 24, 48, 96, 192 h) after the Foc or mock inoculation treatment, and then samples were quick-frozen in liquid nitrogen and stored at −80 °C until analysis.

### 2.2. Total Protein Extraction and Itraq Analysis

Total proteins from cucumber roots were prepared as described by Xu and associates (2019) with minor modifications [35]. The roots were ground into powder in liquid nitrogen, transferred to chilled acetone containing 10% (*v/v*) trichloroacetic acid, and incubated for 1 h at −20 °C. Then the protein pellet was vacuum-dried and redissolved in lysis buffer containing 100 mM NH_4_HCO_3_ (pH 8), 8 M urea and 0.2% SDS, followed by 5 min of ultrasonication on ice. After centrifugation at 12,000× *g* at 4 °C for 15 min, the supernatant of the lysate was transferred to a clean tube and extracts from each sample were reduced with 10 mM DTT for 1 h at 56 °C, and subsequently alkylated with iodoacetamide for 1 h at room temperature in the dark. Then samples were completely mixed with four volumes of precooled acetone by vortexing and incubated at −20 °C for at least 2 h. Samples were then centrifuged and the precipitate was collected. After washing twice with cold acetone, the pellet (peptide mixture) was dissolved in dissolution buffer containing 0.1 M triethylammonium bicarbonate (pH 8.5) and 6 M urea. NanoDrop (Thermo Scientific, Waltham, MN, USA) was used to quantify the protein concentration, and protein integrity was verified by polyacrylamide gel electrophoresis.

The iTRAQ technique, isobaric tags for absolute and relative quantification, is a method for quantitative analysis of the proteome and is more accurate and easier to use than two-dimensional electrophoresis [36]. We performed iTRAQ analysis of samples, with three biological replicates, using reagents supplied by Novogene (Beijing, China). Samples of 100 mg peptide mixture were labeled using the 8-plex iTRAQ reagent kit (Applied Biosystems, Framingham, MA, USA) following the manufacturer’s protocol. These labeled peptides were fractionated by SCX chromatography using the AKTA Purifier system (GE Healthcare, Chicago, IL, USA).

Then, for high-performance liquid chromatography, each fraction was injected into a nanoscale liquid chromatography system (Shimadzu, Kyoto, Japan) coupled to tandem mass spectrometry (LC–MS/MS) analysis. The retained peptides mixture was loaded onto a reverse phase trap column (Thermo Scientific Acclaim Pep Map 100, 2 cm × 100 µm, nano Viper C18) connected to a C18-reverse phase analytical column (Thermo Scientific Easy Column, 10 cm × 75 µm, 3-μm resin) in buffer A (0.1% formic acid) and separated with a linear gradient of buffer B (84% acetonitrile and 0.1% formic acid) at a flow rate of 300 nL/min controlled by IntelliFlow technology. LC–MS/MS analysis was performed on a Q Exactive mass spectrometer (Thermo Scientific, Waltham, MA, USA) coupled to an Easy nLC (Proxeon Biosystems, now Thermo Fisher Scientific) for 2 h. The mass spectrometer was operated in the positive ion mode. All raw mass spectrometry data have been deposited in the iProX Consortium (an official member of ProteomeXchange Consortium) with the dataset identifier IPX0003610001.

### 2.3. Identification of Proteins and Bioinformatic Analysis

All raw files were further analyzed using Protein Pilot software v.5.0 (AB Sciex) with the Paragon algorithm against the cucumber (Chinese Long) genome assembly v.3.0 (http://cucurbitgenomics.org, accessed on 5 November 2021).

Comparing the protein abundance levels between any two groups, when the change in protein abundance is ≥1.2-fold or ≤0.83-fold, the *p*-value is less than 0.05; significant differences in protein expression between treatments were defined as differentially regulated proteins (DRPs) [37]. The UniProt database (http://www.uniprot.org, accessed on 5 November 2021) [38] and Blast2GO (v. 2.7.2) [39] were used for GO terms in classification of DRPs. Enriched GO terms were identified with Fisher’s exact test and a hypergeometric distribution test cut-off of 0.05. Information on the biological pathways of the DRPs was obtained from the Kyoto Encyclopedia of Genes and Genomes (KEGG) pathways database [40]. Selected DRPs were used to identify overlapping categories and generate Venn diagrams (http://bioinformatics.psb.ugent.be/webtools/Venn/, accessed on 5 November 2021). Visualization of significant differences in protein expression between different groups was performed by the Toolbox for Biologist (TBtools) software.

### 2.4. RNA Isolation and Expression Pattern Analysis

Total RNA of cucumber roots, collected at various time points (0, 24, 48, 96, and 192 h) after inoculation with Foc, was isolated using a TaKaRa MiniBEST Plant RNA Extraction Kit (TaKaRa, Dalian, China). The RNA quantity and quality were evaluated with a NanoDrop 2000 spectrophotometer (IMPLEN, Calabasas, CA, USA), after which cDNA was synthesized using the Prime Script Reverse Transcriptase Kit (Takara, Dalian, China) for quantitative real time reverse transcription PCR (qRT-PCR) analysis. The AceQ SYBR Green Master (Vazyme, Nanjing, China) was used in accordance with the manufacturer’s instructions, and PCR amplification was performed using an Iqtm5 Multicolor qPCR detection system (BioRad, USA). The ΔΔCt was the Ct value of the candidate gene minus the Ct values of the reference gene between the treatment samples and controls, respectively. The relative expression was calculated according to the 2^−ΔΔCt^ method, and the standard deviation was calculated using three biological replicates. The cucumber tubulin α chain gene (*CsaV3_4G000060*) was used as the reference for qRT-PCR, and the corresponding primers for the genes encoding the candidate proteins are shown in Appendix A.

### 2.5. Statistical Analysis

Results were analyzed using SPSS v. 20.0 statistical software, and significant differences between groups were identified using Student’s t-tests. Significant differences were represented by * for *p* ≤ 0.05, ** for *p* ≤ 0.01.

## 3. Results

### 3.1. Symptoms of Foc Infection on Rijiecheng and Superina Roots

Our previous studies to identify germplasm resources confirmed that the seedlings of cucumber cultivars Superina and Rijiecheng exhibited highly susceptible and resistant phenotypes, respectively, after Foc inoculation [19]. In further investigations of the difference in symptoms between the resistant and susceptible lines, we found that Rijiecheng (resistant to Foc, R) showed healthy growth, whereas Superina (susceptable to Foc, S) showed obviously wilting leaves and the plants typically died about 10 d after Foc inoculation (Figure 1A). We also used the Foc fused with GFP green fluorescent protein (Foc-GFP) to inoculate Rijiecheng and Superina roots and found a markedly larger number of mycelia on the surface of Superina roots relative to Rijiecheng by 96 h after Foc-GFP inoculation (Figure 1B), suggested that Superina was highly susceptible to cucumber FW, consistent with the disease symptoms.

### 3.2. Identification of the DRPs in Rijiecheng and Superina Roots by iTRAQ

In susceptible host plants, typically, susceptibility (S) genes assist pathogens in host recognition and penetration, and negatively regulate the plant immune signals. Thus, disrupting S genes might interfere with the compatibility between the host and pathogen, and consequently provide broad-spectrum and durable disease resistance [20]. To better identify the S genes underlying the difference in resistance between the two genotypes after Foc infection, we performed an iTRAQ proteomic analysis of Rijiecheng and Superina roots harvested at time points up to 96 h after Foc inoculation. Using Proteome Discoverer 2.2 software, we identified 6558 proteins in the twelve samples. Principal component analysis (PCA) of these samples was performed to distinguish the variation in protein expression of the two cultivars (Figure 2A). Pairwise comparisons of proteins with *p*-values < 0.05 and changes in abundance ≥1.2 or ≤0.83-fold were defined as DRPs. In total, we identified 243 and 210 DRPs by comparing the Foc-infected and mock-inoculated Superina and Rijiecheng seedlings, respectively (Figure 2B, Appendix A), indicating that these proteins might play important roles in cucumber FW defense.

### 3.3. Comparing the Expression of Proteins between the Resistant and Susceptible Lines

We further identified 578 DRPs by comparing the expression of proteins in Superina and Rijiecheng. Among them, we found that 304 proteins had higher expression levels in Superina, whereas 274 proteins were predominantly expressed in Rijiecheng (Figure 3A). To gain insights into the functions of these proteins, the DRPs were analyzed and classified according to the KEGG enrichment pathways. The functional descriptions were mainly associated with the MAPK signals, phenylpropanoid biosynthesis, peroxisome, monoterpenoid biosynthesis, flavonoid biosynthesis, stilbenoid biosynthesis and plant hormone signal transduction (Figure 3B). These findings indicated that these DRPs shared diverse but overlapping functions and might be responsible for FW defense in cucumber.

### 3.4. Profile of Proteins Altered in Cucumber Inoculated with Foc

We further selected 77 DRPs that were predominantly expressed in Superina but also significantly induced by Foc infection, as shown in a Venn diagram (Figure 4A). We further found that 32 DRPs were clearly up-regulated in Superina after Foc inoculation. The expression profiles confirmed that these proteins had higher expression in Superina and were induced by Foc (Figure 4B). Furthermore, the functional annotation of these DRPs according to the KEGG pathway enrichment analysis (Appendix A), indicated that these candidate DRPs might participate in determining susceptibility to FW in Superina.

### 3.5. Validation of Genes Encoding DRPs by qRT-PCR

To further ascertain the functions of candidate DRPs in cucumber defenses against Foc, we selected 16 corresponding genes encoding DRPs with higher expression values to analyze the gene expression patterns in Rijiecheng and Superina after Foc inoculation, using qRT-PCR. We found these genes were up-regulated in the two cucumber lines after Foc infection, relative to the mock-inoculated controls. The expression levels of these genes were induced by Foc and reached peak expression levels at 48 h or 96 h in Rijiecheng (Appendix A), whereas in the Foc-inoculated Superina seedlings, these genes were significantly induced and more quickly reached peak expression values at 24 h (Figure 5A). Furthermore, comparative expression analysis of these genes revealed that the expression levels of four genes were higher in Superina than in Rijiecheng (Figure 5B), namely *TMEM115* (*CsaV3_5G025750*, encoding a transmembrane protein), *TET8* (*CsaV3_2G007840*, encoding a tetraspanin), *TPS10* (*CsaV3_2G017980*, encoding a terpene synthase), and *MGT2* (*CsaV3_7G006660*, encoding a glycosyltransferase), suggesting that these genes might play important roles in the regulation of cucumber susceptibility to *FW*.

## 4. Discussion

Cucumber *FW* is a highly destructive vascular disease and extremely difficult to control because of the limited available natural resistance resources, and also, changes in the pathogenicity and re-instigating the disease cycle [8,19,41]. To date, the related mechanisms of *FW* suppression are largely unknown. There is an urgent need to further investigate host plant–pathogen interaction mechanisms and adopt innovative strategies to produce durable disease-resistant and high-yield crop varieties.

Susceptibility (S) genes are required for successful pathogen infection and penetration, and negatively regulate the host plant immune signals, and thus are considered to play essential roles in plant–pathogen interactions [42,43]. Hence, disrupting S genes might interfere with the compatibility between the host plant and pathogen, which is implicated in prolonged or constitutive defense responses, and consequently might lead to developing broad-spectrum and durable disease resistance.

Several cultivars have been successfully generated and commercialized worldwide [44]. For instance, *Mlo* acts as a typical PM-susceptibility gene, and disruption of *Mlo* has been found in many plant species, and significantly improved durable pathogen-resistance and enhanced yield and quality [45]. Likely, the protein eIF4E, also known as cap-binding protein, is essential for the cellular infection cycle and interaction with viruses, and thus breaking this interaction can confer immunity against viruses in various host plants [46]. Recently, two independent studies have shown that disrupting eIF4Es using the CRISPR-Cas9 system can trigger resistance to several viruses in *Arabidopsis* [33] and cucumber [34]. Also, S genes have been utilized to trigger immunity against various bacterial pathogens. Rice *OsSWEET14* is significantly induced by *Xanthomonas oryzae pv. oryzae* (*Xoo*), and diverts host sugars from the plant cell to the apoplast to supply the pathogen’s nutritional needs [47]. Consistently, breaking this interaction triggers immunity against bacterial blight [48]. Two independent studies have used genome editing techniques to target *OsSWEET* and conferred significant resistance against *Xoo* [49,50]. Taken together, these findings showed that S-gene-mediated resistance generated using genome editing techniques has been successfully applied in different host plants for defense against fungi, bacteria, or viruses.

Future large-scale transcriptome or proteomic analysis of compatible as well as incompatible plant–pathogen interactions, followed by comparative co-expression network analyses, might discover additional novel, nutritional or immunity-related S genes in plants [51,52,53,54]. However, to date, the S genes have not been systematically analyzed in cucumber or effectively utilized for *FW* resistance. Here, we focused on identification of the important S genes in cucumber through comparative proteomic analysis between cultivars Rijiecheng and Superina, which exhibited resistance and high susceptibility to FW, respectively (Figure 1, Figure 3 and Figure 4). On the basis of the expression analysis, we further identified that the genes *TMEM115*, *TET8*, *TPS10*, and *MGT2* were significantly induced in both Rijiecheng and Superina but induced to a greater extent in Superina than in Rijiecheng, suggesting that these candidate gene might relate to the *FW*-susceptibility of Superina (Figure 5). Usually, the S genes are induced and targeted by infected pathogens to negatively regulate host resistance [20]. *Mlo* encodes a transmembrane protein and act as a one of the best-known S genes to use for durable pathogen-resistance programs [27]. The *AtRTP5*, which encodes a WD40 repeat domain-containing protein and is observed at the plasma membrane, negatively regulates plant resistance to *Phytophthora* pathogens in *Arabidopsis thaliana* [55]. The *NAC with*
*transmembrane*
*motif1-like9 (NTL9)* can interact with *Crowded nuclei* (*CRWN1*) to negatively regulate plant immunity through inhibiting the *Pathogenesis-related1 (PR1)* expression in the *A. thaliana* [56]. Furthermore, the glycosyltransferase UGT76B1 can modulate N-hydroxy-pipecolic acid homeostasis to negatively affect plant plant immunity, and knock-out mutant lines of UGT76B1 have constitutive defense response [57]. Likely, in the cucumber, we found that *TMEM115*, *TET8*, *TPS10*, and *MGT2* were significantly induced by the Foc with high expression level in the high-susceptibility cultivar and possessed multiple transmembrane domains, and speculated that they might act as important genes to negatively regulate cucumber *FW* defense. Although the genetic basis for *FW* defense mechanisms remains unclear, our comparative analyses have created a small pool of candidates for further study. This result indicated the importance of these candidate genes in playing negative roles in *FW* defense and should provide effective *FW*-susceptibility gene resources for improving cucumber *FW* resistance. With the availability of efficient multiplex transgene-free systems using new genome editing tools, the introduction of broad-spectrum *FW* resistance, for example, by targeting multiple S genes simultaneously, has also become practical.

## 5. Conclusions

We performed a comparative proteomic analysis between cucumber cultivars Rijiecheng and Superina, which exhibited resistance and high susceptibility to FW, respectively, to identify the important S genes in cucumber defense. We further found that 32 differentially regulated proteins (DRPs) were predominantly expressed in Superina and significantly up-regulated after Foc inoculation. On the basis of the expression verification, we further identified that the genes *TMEM115* (*CsaV3_5G025750*), *TET8* (*CsaV3_2G007840*), *TPS10* (*CsaV3_2G017980*), and *MGT2* (*CsaV3_7G006660*) were significantly induced in both Rijiecheng and Superina but induced to a greater extent in Superina than in Rijiecheng, suggesting that these candidate gene might relate to the *FW*-susceptibility of Superina. We speculate that these four candidate genes may act as negative regulators of *FW* resistance in cucumber, and further research is required to define the precise underlying molecular mechanisms. These data provide important information to highlight the molecular processes of disease susceptibility in cucumber and facilitate further studies in cucumber breeding for disease resistance.

## Figures and Tables

**Figure 1 genes-12-01781-f001:**
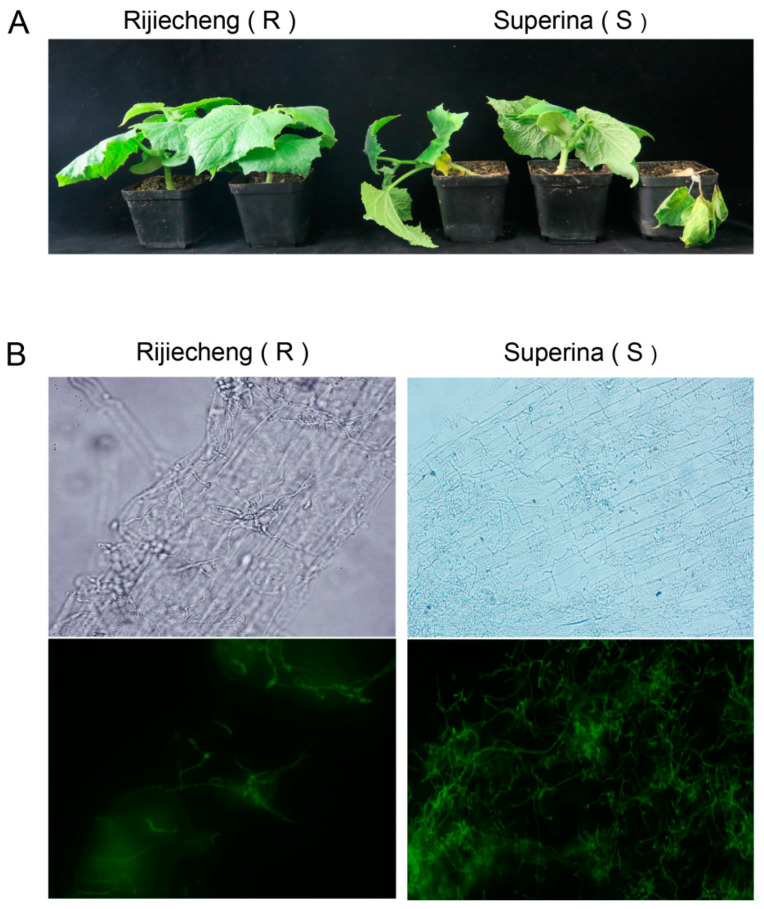
Phenotype of cucumber cultivars Rijiecheng and Superina infected with Foc. (**A**) Typical symptoms on cucumber 10 d after infection with the Foc pathogen; (**B**) Phenotypes of Rijiecheng and Superina roots by fluorescence microscopy 96 h after Foc-GFP inoculation. Left: Rijiecheng (R); Right: Superina (S).

**Figure 2 genes-12-01781-f002:**
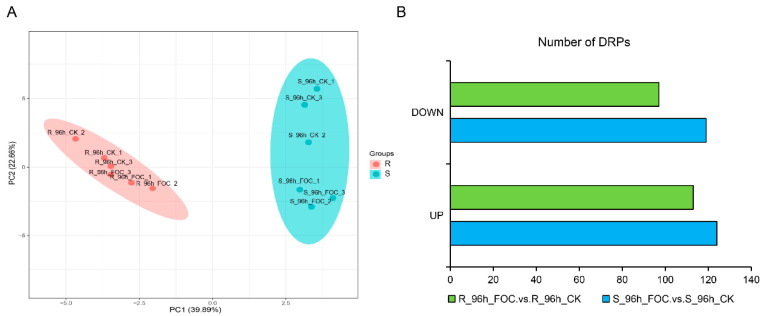
Identification of differentially regulated proteins (DRPs) in Rijiecheng and Superina (**A**) Principal component analysis of iTRAQ samples. Each point represents the whole-protein profile of a single biological sample. (**B**) Number of DRPs in Rijiecheng and Superina roots 96 h after infection with Foc. Down and Up indicate down- and up-regulated expression patterns relative to the control; the horizontal axis represents the number of DRPs. R: Rijiecheng; S: Superina.

**Figure 3 genes-12-01781-f003:**
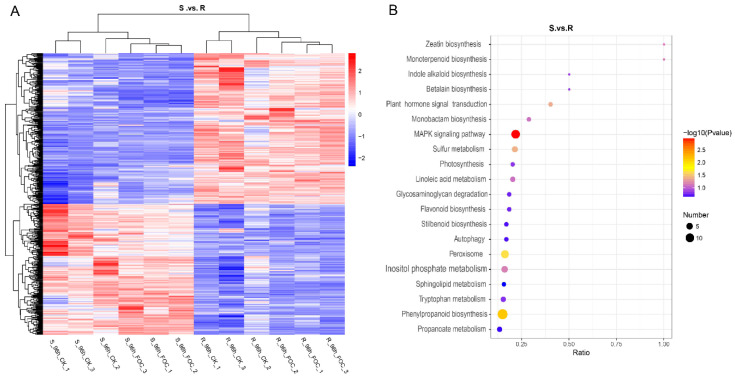
Functional classification of differentially regulated proteins (DRPs) between two cucumber lines (**A**) Comparative analysis of expression patterns of DRPs in the two cucumber lines. Up- and down-regulated proteins in the roots are colored red and blue, respectively. (**B**) KEGG pathway enrichment of DRPs between two cucumber lines. The size of the bubbles corresponds to different numbers of DRPs, and the ordinate represents different KEGG classifications. The color scale indicates relative gene expression.

**Figure 4 genes-12-01781-f004:**
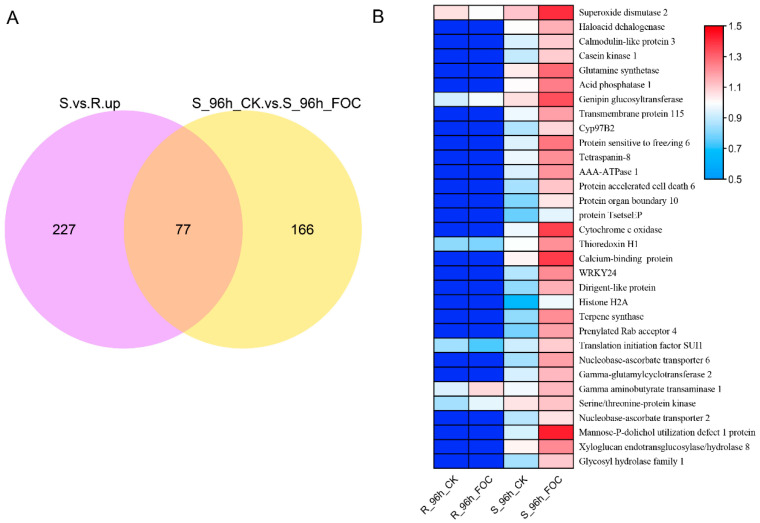
Expression patterns of selected DRPs (**A**) Venn diagram of up-regulated proteins in Superina infected by Foc. (**B**) Expression patterns of candidate DRPs in Rijiecheng and Superina after Foc inoculation. Functional annotation of DRPs is shown at the right of the graph. Proteins more highly or more weakly expressed in the roots are colored red and blue, respectively.

**Figure 5 genes-12-01781-f005:**
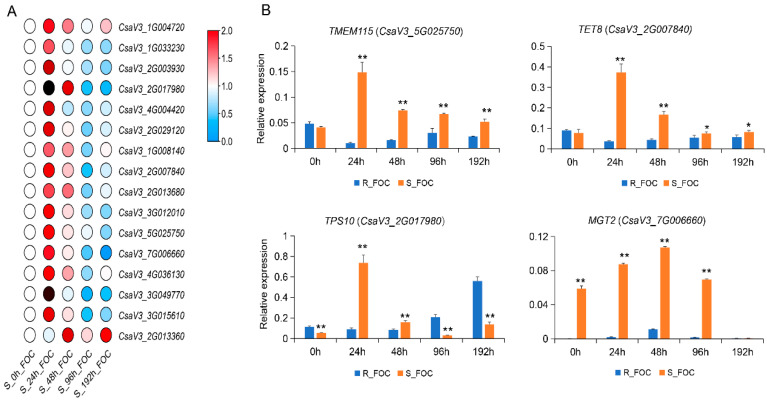
Expression of genes corresponding to candidate DRPs (**A**) qRT-PCR analysis of 16 candidate genes in Superina roots at different time points after inoculation with Foc. The expression values were calculated according to the 2^−ΔΔCt^ method with three biological replicates. The heatmap was generated using TBtools software, and genes more highly or weakly expressed in the roots are shown in red and blue, respectively (scale 0–2 for relative expression). (**B**) Expression levels of four candidate genes in Rijiecheng and Superina after Foc inoculation. The error bars with standard deviations were calculated from three biological replicates. * significantly different at *p* ≤ 0.05; ** significantly different at *p* ≤ 0.01.

## Data Availability

All raw mass spectrometry data associated with this study have been deposited in the iProX Consortium with the dataset identifier IPX0003610001.

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
