# Peer review of "Identification of Susceptibility Genes for Fusarium oxysporum in Cucumber via Comparative Proteomic Analysis"

_genes, 2021, doi:10.3390/genes12111781_

Round 1

Reviewer 1 Report

The work by Chen et al. is an interesting work as it identified the susceptibility proteins and pathways for Fusarium oxysporum in cucumber via comparative proteomic analysis. This information could be used for resistance breeding. 

English should be checked by a native speaker. There are some mistakes, such as in the introduction,

Line 29 Cucumber (Cucumis sativus L.) is one of the most economically important vegetable crops and is cultivated worldwide.

32: When the Foc pathogen successfully invades the plant, it results in wilting of wilts the leaves or even the entire plant.

34: it is usually considered as the most serious severe biotic factor limiting global cucumber production

39: However, available natural resistance resources for FW are very limited, meaning that the breeding of resistant cultivars through conventional approaches is still a huge considerable challenge.

57: shown to significantly enhance PM resistance in wheat significantly.

67:Currently, genetic resources for FW resistance in cucumber are very limited and FW has not been effectively controlled by the breeding of resistant varieties using conventional approaches.

70: Thus, a cost-effective and environmentally friendly strategy for limiting diseases is disrupting disrupt S genes to obtain sustainably disease-resistant cultivars sustainably.

For the introduction, I expect to learn more about Fusarium oxysporum in cucumber, including how they induce disease and the known QTLs in resistance breeding. The genes you mentioned in the introduction do not relate to fusarium wilt very well.

Materials and Methods:

Line 96: I think it should be 10^6 spores/ml.

As I know, the main symptom of fusarium wilt is the wilting of the entire plant, which could be caused by a fungal toxin(s) or blockage of water transport. The experiments only used roots for analysis. It’s better to clear the reason.

Discussion

The discussion is very similar to the introduction. The results only suggested the four selected genes related to fusarium wilt. We do not know if they indeed cause susceptibility or not. The authors should discuss the gene function and proposed mechanisms involved in the disease infection. Also, it’s better to have further data supporting those selected genes are really S genes.

Overall, it should  do minor modification.

Author Response

Dear Reviewer:

  Thank you very much for giving us an opportunity to revise our manuscript. We greatly appreciate you for your valuable comments and constructive suggestions to improve the manuscript. According to your suggestions, the English text of a draft of this manuscript have edited by a PhD, Huw Tyson, from Liwen Bianji, Edanz Editing China (www.liwenbianji.cn/ac), and we tried our best to revise the manuscript, reorganize our results and discussions, which make the theme of paper clearer and read more smoothly. Finally, we answered all comments from the reviewers in order, by listing all suggestions followed by our indications of how we have addressed these concerns.

We hope that you find this revised manuscript satisfactory for publication in Genes journal. Thank you for your time and consideration.

Sincerely yours,

Responses to the Reviewer (Q, question; A, answer):

  1. Q:Line 29 Cucumber (Cucumis sativus L.) is one of the most economically important vegetable crops and is cultivated worldwide

A: Thank you very much for your careful examination. In the revised manuscript (line 28 to 29), we had amended this sentence.

  1. Q:Line 32: When the Foc pathogen successfully invades the plant, it results in wilting of wilts the leaves or even the entire plant

A: Thank you so much for your comment. In the revised manuscript (line 32), we had amended this sentence. 

  1. Q: Line 34: it is usually considered as the most serious severe biotic factor limiting global cucumber production

A: Many thanks for your careful examination. In the revised manuscript (line 34), we had amended this sentence. 

  1. Q:Line 39: However, available natural resistance resources for FW are very limited, meaning that the breeding of resistant cultivars through conventional approaches is still a huge considerable challenge.

A: Thank you so much for your suggestion. In the revised manuscript (line 53 to 55), we had amended the descriptions.

  1. Q:Line 57: shown to significantly enhance PM resistance in wheat significantly.

A: Thank you very much for your careful examinations. In the revised manuscript (line 71), we had amended the descriptions.

  1. Q: Line 67:Currently, genetic resources for FW resistance in cucumber are very limited and FW has not been effectively controlled by the breeding of resistant varieties using conventional approaches.

A: Many thanks for your comments. In the revised manuscript (line 81 to 83), we had amended the sentence.

  1. Q: Line 70: Thus, a cost-effective and environmentally friendly strategy for limiting diseases is disrupting disrupt S genes to obtain sustainably disease-resistant cultivars sustainably.

A: Many thanks for your comments. In the revised manuscript (line 85 to 87), we had amended the sentence.

  1. Q: For the introduction, I expect to learn more about Fusarium oxysporum in cucumber, including how they induce disease and the known QTLs in resistance breeding. The genes you mentioned in the introduction do not relate to fusarium wilt very well.

A: Many thanks for your suggestion. In the revised manuscript (line 43 to 52), we had added descriptions to enrich this topic (cucumber and Fusarium wilt ) including identification resistant genes or effect quantitative trait loci (QTLs) from the resistant breeding resources.

  1. Q: Line 96: I think it should be 10^6 spores/ml.

A: Many thanks for your comments. In the revised manuscript (line 115), we had amended the description.

  1. Q: As I know, the main symptom of fusarium wilt is the wilting of the entire plant, which could be caused by a fungal toxin(s) or blockage of water transport. The experiments only used roots for analysis. It’s better to clear the reason.

A: Many thanks for your suggestion. The Foc acted as a typical soil-borne and destructive fungal disease, which firstly infected the roots of cucumber in the growth process and induced the responsive genes to defense FW. Hence, we selected the roots to analyze and identify the FW-responsive genes. In the revised manuscript (line 109 to 111), we had added the description of reason.

  1. Q:The discussion is very similar to the introduction. The results only suggested the four selected genes related to fusarium wilt. We do not know if they indeed cause susceptibility or not. The authors should discuss the gene function and proposed mechanisms involved in the disease infection. Also, it’s better to have further data supporting those selected genes are really S genes.

A: Thank you very much for your comments and suggestions. In the revised manuscript, we had rewritten the discussion and added the detailed descriptions to discuss the gene functions and proposed mechanisms in different plants (line 329 to 343). Also, on the basis of comparative proteomics and expression verification in this study, we found that four candidate genes were predominantly expressed in the high-susceptibility cultivar after Foc infection. Taken together, we speculate that these four candidate genes may act as negative regulators of FW resistance in cucumber, and further research is required to define the precise underlying molecular mechanisms.

Reviewer 2 Report

The authors investigated one of the most economically important vegetable  cucumber, and pathogen, that damaged these vegetables Fusarium wilt.  They identified differentially regulated proteins in two cucumber cultivars and found candidate genes that might act as negative regulators of cucumber resistance in Fusarium wilt.

Extensive research had been conducted. However I have a few comments for introduction and discussion.  At first `I missed a more detailed description of what has already been done on this topic (about cucumber and Fusarium wilt), especially papers that written in 2015-2021 y. for the readers are actual what works are done yet.  For example: Gu Z, Wang M, Wang Y, Zhu L, Mur LA, Hu J, Guo S. Nitrate stabilizes the rhizospheric fungal community to suppress Fusarium wilt disease in cucumber. Molecular Plant-Microbe Interactions. 2020 Apr 23;33(4):590-9.Zhou, J., Wang, M., Sun, Y., Gu, Z., Wang, R., Saydin, A., Shen, Q. and Guo, S., 2017. Nitrate increased cucumber tolerance to Fusarium wilt by regulating fungal toxin production and distribution. Toxins9(3), p.100.Gao, X., Li, K., Ma, Z., Zou, H., Jin, H. and Wang, J., 2020. Cucumber Fusarium wilt resistance induced by intercropping with celery differs from that induced by the cucumber genotype and is related to sulfur-containing allelochemicals. Scientia Horticulturae271, p.109475. and other papers. 

Second, you don't have to write your results in the introduction (from line 72). You must to write why you decided to do these experiment, what your goals and hypothesis. 

Methods. Lines 102-115. Maybe you can point out the reference from where these protocols are taken?

In the Discussions I miss a detailed discussion of your results and comparing with other papers. Lines 262-296 are more suitable for the introduction and some lines are the same as in the introduction. You must rewrite Discussions and write separate chapter of Conclusions.

Author Response

Dear Reviewer:

  Thank you very much for giving us an opportunity to revise our manuscript. We greatly appreciate you for your valuable comments and constructive suggestions to improve the manuscript. According to your suggestions, the English text of a draft of this manuscript have edited by a PhD, Huw Tyson, from Liwen Bianji, Edanz Editing China (www.liwenbianji.cn/ac), and we tried our best to revise the manuscript, reorganize our results and discussions, which make the theme of paper clearer and read more smoothly. Finally, we answered all comments from the reviewers in order, by listing all suggestions followed by our indications of how we have addressed these concerns.

We hope that you find this revised manuscript satisfactory for publication in Genes journal. Thank you for your time and consideration.

Sincerely yours,

Responses to the Reviewer (Q, question; A, answer):

  1. Q:Extensive research had been conducted. However I have a few comments for introduction and discussion.  At first `I missed a more detailed description of what has already been done on this topic (about cucumber and Fusarium wilt), especially papers that written in 2015-2021 y. for the readers are actual what works are done yet.  For example: Gu Z, Wang M, Wang Y, Zhu L, Mur LA, Hu J, Guo S. Nitrate stabilizes the rhizospheric fungal community to suppress Fusarium wilt disease in cucumber. Molecular Plant-Microbe Interactions. 2020 Apr 23;33(4):590-9. Zhou, J., Wang, M., Sun, Y., Gu, Z., Wang, R., Saydin, A., Shen, Q. and Guo, S., 2017. Nitrate increased cucumber tolerance to Fusarium wilt by regulating fungal toxin production and distribution. Toxins, 9(3), p.100.Gao, X., Li, K., Ma, Z., Zou, H., Jin, H. and Wang, J., 2020. Cucumber Fusarium wilt resistance induced by intercropping with celery differs from that induced by the cucumber genotype and is related to sulfur-containing allelochemicals. Scientia Horticulturae, 271, p.109475. and other papers.

A: Many thanks for your comments and suggestions. In the revised manuscript (line 38 to 52), we had added descriptions and papers to better enrich this topic (cucumber and Fusarium wilt ) including the grafting with rootstock, chemical compounds, biocontrol via other beneficial fungi or bacteria, and identification resistant genes or effect quantitative trait loci (QTLs) from the resistant breeding resources.

  1. Q:Second, you don't have to write your results in the introduction (from line 72). You must to write why you decided to do these experiment, what your goals and hypothesis

A: Thank you so much for your suggestions. In the revised manuscript (line 82 to 100), we had amended and rewritten the paragraph to better state the purpose/objectives of this research.

  1. Q: Lines 102-115. Maybe you can point out the reference from where these protocols are taken?

A: Many thanks for your suggestions. In the revised manuscript (line 121 to 125), we had added the reference of the protocol.

  1. Q:In the Discussions I miss a detailed discussion of your results and comparing with other papers. Lines 262-296 are more suitable for the introduction and some lines are the same as in the introduction. You must rewrite Discussions and write separate chapter of Conclusions.

A: Thank you so much for your comments and suggestions. In the revised manuscript (line 288 to 350), had removed some repeated descriptions, and rewritten the discussions to further describe the gene functions and proposed mechanisms in different plants, and connected the innovative literature to better improve our manuscript. Also, we had added a separate chapter of conclusions (line 353 to 367) to further summarize our results.